# Effects of Subjective Health Perception on Health Behavior and Cardiovascular Disease Risk Factors in Patients with Prediabetes and Diabetes

**DOI:** 10.3390/ijerph19137900

**Published:** 2022-06-28

**Authors:** Sungjung Kwak, Yoonmi Lee, Seunghui Baek, Jieun Shin

**Affiliations:** 1Robotic Surgery Center, Konyang University Hospital, Daejeon 35365, Korea; sungjungee@gmail.com; 2Department of Health Exercise Management, Sungshin Women’s University, Seoul 02844, Korea; 9988gogo@sungshin.ac.kr; 3Department of Biomedical Informatics, College of Medicine, Konyang University, Daejeon 35365, Korea

**Keywords:** diabetes mellitus, prediabetic, health statues, CVD, health behavior, NHANES

## Abstract

The purpose of this study is to confirm the health behavior performance rate and cardiovascular disease-related indicators according to the subjective health perception of prediabetic and diabetic patients using the 2016–2019 National Health and Nutrition Examination Survey (NHANES) data. This study classified hemoglobin A1c ≥ 6.5% as diabetes and 5.7–6.4% prediabetes among 2485 adults over 40 years of age among the data from the National Health and Nutrition Examination Survey. In addition, subjective health perception was divided into ‘good’ and ‘bad’ and then cross-classified into four groups (Good/PDM, Bad/PDM, Good/DM, and Bad/DM) to analyze the differences between the four groups. All statistical analyses were performed using SAS 9.4 (SAS Institute Inc., Cary, NC, USA), and complex sample analysis was performed using weights according to the KNHANES raw data usage guidelines from the Korea Centers for Disease Control and Prevention. The rate of subjective health awareness was higher in men than in women in both prediabetic- and diabetic-stage subjects and adults in the prediabetic stage were higher than in the elderly. The better the subjective health status, the higher the ratio of normal weight, proper sleep time, exercise, and eating out among health-related behaviors. As a result of analyzing blood pressure and blood indices related to the cardiovascular disease risk (Framingham Risk Score), in all indicators except blood pressure, the better the subjective health status and the lower the risk of cardiovascular disease. As a result, for disease prevention and continuous management through healthy behaviors in prediabetic and diabetic patients, it is necessary to improve the positive subjective perception of health.

## 1. Introduction

Diabetes is one of the diseases with the most rapid increases in prevalence worldwide; approximately every 10 s, three individuals develop diabetes. The number of patients diagnosed with diabetes is projected to increase to approximately 59 million by 2035 [1]. In South Korea, approximately one in seven adults aged 30 years or older and approximately three out of ten elderly people aged 65 years or older are currently diagnosed with diabetes [2]. However, the proportion of patients achieving an HbA1c of <6.5% stands at 28.3%, and the proportion of those requiring active treatment (HbA1c of ≥8%) is at 19%. This indicates a lack of or inadequate diabetes management and suggests that the existing intervention programs are not sufficient to induce and sustain behavioral changes in patients diagnosed with the disease.

As diabetes is associated with a high incidence of fatal complications, such as cardiovascular disease and diabetic kidney disease, as well as having a high mortality rate, the effective health management of diabetes is imperative [2,3]. Diabetes is a particularly strong independent risk factor for cardiovascular disease [4], with men with diabetes having a 2–3 times increased risk and women a 3–4 times increased risk of cardiovascular disease compared to that in the general healthy population. Patients with diabetes also develop cardiovascular disease 15 years earlier on average [5], and approximately 20% of patients with diabetes die due to cardiovascular complications [2]. Health management is, therefore, essential for patients as well as for individuals at a high risk who have not yet developed diabetes.

Health management is affected by an individual’s subjective health status, which represents the self-evaluation of one’s overall health [6]. This subjective perception of one’s health determines attitudes that do or do not lead to health behaviors [7]. It has been reported that those who perceive themselves as being in good health practice more health behaviors, indicating that subjective health status is an important factor in inducing individual behavior changes [8,9]. In patients with diabetes in particular, subjective health status is considered a reliable measure of an individuals’ health [10], and, in Europe, patients with diabetes who perceive their health status as bad were reported to have a higher mortality [11]. 

Studies investigating subjective health perception among patients with diabetes to date have focused on quality of life [12,13], depression [14], influential factors [15], mediating effects [16], HbA1c [17], suicidal behavior [18], and mental health [19]. However, no study so far has reported on health behavior compliance and cardiovascular indices related to subjective health perception in patients with diabetes.

The aim of this study was, therefore, to investigate health behavior compliance and the relationship between cardiovascular disease-related indices and subjective health perception in patients with prediabetes and diabetes. We suggest that our results be used as the basis for the development of intervention strategies to reduce the prevalence of diabetes and the incidence of complications related to the disease.

## 2. Methods

### 2.1. Study Design

This descriptive research study used a secondary analysis of the raw data from the 7th (2016–2018) and 8th (2019) Korea National Health and Nutrition Examination Surveys to investigate cardiovascular disease-related risk factors as well as the effects of subjective health perception on health behavior in patients with prediabetes and diabetes.

### 2.2. Participants and Data Collection

We selected among a total of 32,379 people who participated in the 7th (2016–2018) and 8th (2019) National Health and Nutrition Survey, 19,235 people over the age of 40. The following respondents were then excluded: (1) those (*n* = 6908) for whom HbA1c levels were not available or those who had reported normal HbA1c levels (<5.6); (2) those who had not responded to the subjective health status item or had responded with “normal” (*n* = 23,310); (3) those who were pregnant or breastfeeding at the time of the survey (*n* = 1); (4) those undergoing dietary therapy due to disease at the time of the survey (*n* = 594); (5) those with cancer (*n* = 390); and (6) those who had not responded to the health behavior item (*n* = 1809). A total of 2,485 individuals were finally selected as participants for this study and were classified into a group with diabetes (HbA1c ≥ 6.5%) and a group with prediabetes (HbA1c 5.7–6.4%) [20]. Subsequently, they were cross-classified according to their subjective health status into those who perceived their health as “good” and those who considered themselves as being in “bad” health and divided into the following four groups: prediabetes stage with good self-perceived health status (“Good/PDM”, 1123 individuals), prediabetes stage with bad self-perceived health status (“Bad/PDM”, 904), diabetes stage with good self-perceived health status (“Good/DM”, 151), and diabetes stage with bad self-perceived health status (“Bad/DM”, 307).

### 2.3. Instruments

#### 2.3.1. Subjective Health Status

Subjective health status in this study was rated on a 5-point Likert scale, with the levels “very good”, “good”, “normal”, “bad”, and “very bad”. Respondents had been asked the question “How do you usually think about your health status?” Responses were classified as good (“very good” and “good”), bad (“bad” and “very bad”), and “normal”.

#### 2.3.2. Health-Related Behaviors

We used the seven health-related behaviors referred to as the “Alameda 7”, (non-smoking, moderate drinking, adequate sleep, maintaining a desirable weight, exercising, eating breakfast, and avoiding snacks), but replaced “avoiding snacks” with “eating out rarely”. Non-smoking was defined as having never smoked in one’s lifetime or having smoked in the past but not currently. Moderate drinking was defined as not drinking throughout one’s lifetime or drinking less than twice a week within the past year. A desirable weight was defined as having a body mass index (BMI) of ≥18.5 kg/m^2^ to 25 kg/m^2^ based on the World Health Organization BMI classification for the Asian Population [21]. Adequate sleep was defined as sleeping 7 to 8 h per night. Exercising was defined as a health behavior if a person reported exercising for 150 min or longer per week [21]. Having breakfast was defined as a health behavior if a person had breakfast 5–7 times a week. Eating out rarely was defined as a health behavior if a person reported eating out less than 5 times a week.

#### 2.3.3. Cardiovascular Disease Risk-Related Indicators

The Framingham Risk Score (FRS) was developed based on data obtained from the long-term Framingham Heart Study conducted in the USA and is used to estimate the 10-year cardiovascular risk of individuals aged 30–74 years without cardiovascular disease by scoring for age, total cholesterol, high-density cholesterol (HDL), blood pressure levels, smoking status, and the presence or absence of diabetes. A higher score indicates a greater risk of cardiovascular disease [22].

#### 2.3.4. Body and Blood Indices

Systolic blood pressure (SBP), diastolic blood pressure (DBP), triglyceride (TG), HDL, cholesterol, and fasting blood glucose levels were examined according to the goal of integrated management of patients with diabetes as recommended in the Clinical Practice Guidelines for Diabetes 2021 [2].

## 3. Data Analysis

Statistical analyses were performed using SAS ver. 9.4 (SAS Institute Inc, Cary, NC, USA). Complex sample data analysis was performed using weights in accordance with the Korea Disease Control and Prevention Agency’s guidelines for the use of raw data from the Korea National Health and Nutrition Examination Survey. The participants were divided into four groups by cross-classifying subjective health status (good/bad) and diabetes status (prediabetes/diabetes), and the general characteristics of the participants were analyzed using a crossover analysis. Then, a crossover analysis was performed to compare health behaviors between groups, and a general linear model analysis was conducted to determine differences in blood indices between groups.

### Ethical Considerations

This study represents a secondary analysis of national survey data available through the Korea National Health and Nutrition Examination Survey website and was conducted after receiving approval from the institutional review board of S University (SSWUIRB-2022-024). Because the raw data analyzed for this study contain no personally identifiable information, anonymity and confidentiality were guaranteed.

## 4. Results

### 4.1. General Characteristics of the Participants

The proportion of men was highest in the Good/PDM group (53.1%), followed by the Bad/PDM (28.6%), Bad/DM (10.5%), and Good/DM (7.8%) groups, while the proportion of women was highest in the Good/PDM group (43.5%), followed by the Bad/PDM (39.8%), Bad/DM (12.1%), and Good/DM (4.6%) groups. In both disease groups, that is, among individuals with prediabetes and as well as those with diabetes, more men than women perceived their subjective health status as good, while more women than men perceived their health status as bad.

Body weight was highest in the Good/DM group (mean = 68.7 kg), followed by the Bad/DM group (67.3 kg), the Good/PDM group (65.3 kg), and then the Bad/PDM group (63.4 kg). BMI was highest in the Bad/DM group (26.4 kg/m^2^), followed by the Good/DM (26.0 kg/m^2^), Bad/PDM (24.8 kg/m^2^), and Good/PDM (24.3 kg/m^2^) groups. These findings indicate that body weight and BMI were higher in both DM groups overall, but that BMI was higher in those individuals with poor subjective health.

Regarding age, the majority of adults (40–59 year) fell into the Good/PDM category (58.7%), followed by Bad/PDM (27.6%), Bad/DM (11.3%), and Good/DM (6.1%). For elderly people (>60 year), the percentage of those in the Bad/PDM group was higher (40.9%), followed by the Good/PDM (38.1%), Bad/DM (14.3%), and Good/DM (6.7%) groups. Overall, the proportion of individuals with a good subjective health status and prediabetes (Good/PDM) was approximately 20% higher in adults than in the elderly age group. In terms of marital status, the subsample of survey participants who reported having a spouse had the highest proportion of individuals who fell into the Good/PDM category, at 52.9%. A higher household income was also associated with a higher proportion of individuals in the Good/PDM group, as was a higher education level. Among women, the proportion of respondents in the Good/PDM group was higher, at 62.4%, in non-menopausal women (see Table 1).

### 4.2. Health Behaviors

Among the seven assessed health behaviors, the differences between the ratios of individuals reporting a normal weight, an appropriate sleep time, an appropriate exercise time, and a low frequency of eating out were statistically significant among the groups.

The proportion of those with a normal weight was the highest in the Good/PDM group (33.1%), followed by the Bad/PDM (29%), Bad/DM (21.7%), and Good/DM (16.1%) groups. The proportion of those reporting adequate sleep was the highest in the Good/PDM group (32.9%), followed by the Good/DM (31.8%), Good/PDM (28.7%), and Bad/DM (20%) groups. The proportion of those exercising for an adequate time was the highest in the Good/PDM group (47.4%), followed by the Bad/PDM (30.4%), Good/DM (30.4%), and Bad/DM (26.5%) groups. The proportion of those eating out less was the highest in the Bad/PDM group (76.3%), followed by the Bad/DM (74.3%), Good/DM (64.2%), and Good/PDM (58.2%) groups.

These findings indicate that the proportion of individuals with a normal weight or those exercising for an adequate time was higher in the group with prediabetes. The proportion of individuals reporting an adequate sleep time was higher among those with a good subjective health status, while the proportion of those eating out several times per week was higher among those with poor subjective health (see Table 2 & Figure 1).

### 4.3. Cardiovascular Disease-Related Indices

Mean systolic blood pressure was the highest in the Bad/DM group (128.88) and the lowest in the Good/PDM group (120.88). The mean diastolic blood pressure was the highest in the Good/DM group (77.92) and the lowest in the Bad/DM group (75.84). The mean total cholesterol level was the highest in the Good/PDM group (202.61) and the lowest in the Bad/DM group (181.6), while the mean HDL level was the highest in the Good/PDM group (50.18) and the lowest in the Good/DM group (43.87). The mean LDL level was the highest in the Good/PDM group (122.43) and the lowest in the Bad/DM group (105.44). The mean CVD score was the lowest in the Good/PDM group (10.75) and the highest in the Bad/DM group (13.41). The mean triglyceride level was the highest in the Good/DM group (199.56) and the lowest in the Good/PDM group (141.99). The mean fasting blood glucose level was the highest in the Bad/DM group (15.054) and the lowest in the Good/PDM group (100.55) (see Table 3).

## 5. Discussion

One in 10 adults worldwide have diabetes [23], which is associated with an increased risk of diseases, such as cardiovascular disease and mortality [24]. Therefore, there is an emerging need for social efforts to prevent chronic disease early and reduce social costs due to health deterioration. Study results have revealed that glycemic control through various interventions in patients with diabetes increases the quality of life [25] and reduces complications, medical expenses [25,26], and the prevalence of the cardiovascular disease [27].

Based on the results of studies [8,9,15] indicating that a higher perceived subjective health is, in general, associated with better health behavior, we investigated whether such findings can be applied to patients with diabetes as well.

Our findings show that, first, among those with prediabetes and diabetes, men perceive their health as better overall than women and that in those with prediabetes, adults perceived their health as better than elderly people. These results are consistent with the results of a previous study showing that female elderly people perceived their own health to be worse compared to male elderly people in terms of sex-specific subjective health statuses [28]. Our findings are, however, contradictory to the results of a study in which elderly patients with diabetes were reported to have a positive subjective health perception [17] and wherein subjective health perception became more positive with increasing age [29]. Considering the results of other studies indicating that subjective health perception in elderly subjects is affected by life satisfaction and mood state [30] and that an individual’s experience with illness can affect their evaluation of their health status negatively [31], more research is needed to draw clear conclusions. Furthermore, the factors affecting subjective health status and health-related behaviors differ by sex [15,17]. In addition, even when the same health-related behaviors are implemented, the effects on health indicators vary by sex as well [32], and this factor should be given priority in interventions for diabetic patients.

Second, we found that a better subjective health status is associated with a higher proportion of individuals with a normal weight, an adequate sleep time, an adequate exercise time, and eating out only a few times per week. These results are similar to the results of a study revealing that insomnia, depression, and stress affect subjective health status and that those with insomnia in particular, who find it difficult to fall and stay asleep, perceive their subjective health status to be poor [29]. Moreover, other studies have shown that stress-related symptoms impair subjective health perception [32,33]. In addition, those who exercise regularly were reported to feel more subjectively healthy than those who do not [34], and menopausal women with a high participation in exercise were reported to have a good subjective health status and a significantly lower incidence of menopausal symptoms [35]. The results of a meta-analysis of studies regarding various exercise interventions in patients with diabetes indicate that exercise interventions had the most positive effects on blood glucose control in patients with diabetes [36]. These findings suggest that exercise can help improve disease condition, thereby enhancing subjective health perception.

We found that the individuals in the group with a good subjective health reported eating out more often than those in other groups. This suggests that people who tend to be confident about their health may eat out more, and there may, thus, be a difference between what is perceived as socially acceptable for patients with prediabetes and diabetics and what are recommendations by medical practitioners. In addition, we found that individuals in the Bad/DM group reported the lowest percentage of eating out. We assume that this stems from the fact that dietary management is an important factor for glycemic control in diabetic patients and that patients with diabetes refrain from eating out due to limited food choices and the generally high content of sodium and sugar in the available foods.

Most of the assessed health-related behavior variables were found to have a significant effect in the Good/PDM group. This is consistent with the results of studies showing that subjective health perception positively affects health behavior [8,9,15], that the prevalence of diabetes is higher in those with poor than in those with good subjective health [37], that subjective health is a factor affecting the prevalence of diabetes in elderly people [38], and that those who evaluate their own physical, mental, and cognitive functions more positively are more likely to practice health-promoting behaviors [39].

Overall, patients with diabetes tend to perceive their subjective health as poor compared to those in the general population [40]. To sum up the results of this study and put them in context with earlier research, it seems that for patients with diabetes, perceiving their health as better helps their physical health to some extent. Therefore, when developing intervention strategies for patients with diabetes, it is necessary to carefully assess and manage their own perception of their health. In addition, it is necessary to develop strategies that can improve patients’ positive perception of their health, involving their weight, sleep time, and exercise factors that affect subjective health status, and emphasize dietary habit changes with regard to eating out.

Finally, the results of our analysis of blood pressure and blood indices related to cardiovascular disease risk (FRS) reveal that a better subjective health status was associated with a lower risk of developing cardiovascular disease in all indices except for blood pressure. The FRS estimates the 10-year risk of developing coronary heart disease in adults aged 30–74 years according to sex by scoring for age, smoking status, the presence or absence of diabetes, systolic and diastolic blood pressure, total cholesterol, and HDL cholesterol levels [22]. As regards diabetes and smoking, the participants in this study were patients with prediabetes/diabetes, and we assessed their relationship with smoking through their health behaviors. Our results show that the FRS was the lowest in the Good/PDM group. We then compared the relative FRS differences between groups to investigate cardiovascular disease risk according to subjective health status. Since the FRS was measured without the variables diabetes and smoking, we cannot assume that it accurately determined cardiovascular disease risk. However, it does indicate that the subjective health status in patients with prediabetes and diabetes is significantly associated with indices related to cardiovascular disease as a major complication of diabetes and that a better subjective health status is associated with a reduced risk of cardiovascular disease.

Because previous studies regarding cardiovascular disease according to subjective health status are scarce, a direct comparison between the results of this study and other studies cannot be made. However, it has been reported that better objective medical indices in patients with chronic obstructive pulmonary disease are associated with better subjective health [41], that a better subjective health status in patients with type 2 diabetes is associated with lower HbA1c and blood glucose levels [17], and that a better subjective health status is associated with a higher health-related quality of life [42,43], in line with the results of this study. On the other hand, the lower the illness perception, the different from the results of studies [44], in which show a good control of blood sugar. Continuous research on the effects of self-health and the awareness of one’s disease on health management is necessary.

## 6. Conclusions

This study shows that in both patients with prediabetes and those with diabetes, male patients have a better subjective health perception than female patients and that those with a good subjective health perception also practiced more good health behaviors (related to sleep, weight, and exercise). This suggests that enhancing the subjective health perception in patients with prediabetes and diabetes increases the probability of also improving their diabetes. Although various studies regarding methods for improving glycemic control in patients with prediabetes/diabetes have been conducted, most are related to basic diabetes guidelines on dietary control and exercise. For more systematic and efficient managements of patients with prediabetes/diabetes, it is necessary to first identify the subjective health perception at an individualized level and provide customized programs for individual patients that can improve their subjective health perception and be applied to public health system management.

This study is significant in that it investigated the effects of subjective health perception on health behaviors and cardiovascular risk factors in patients with prediabetes/diabetes and provides basic data for the development of programs for improving diabetes. Because our study used a nationally representative large-scale sample in South Korea, it is possible to generalize the results of this study. However, this study has the limitation in that causal relationships cannot be assumed, due to its cross-sectional nature. In addition, because we were not able to include other factors affecting individuals’ behavior such as attitude and values, large-scale studies considering these factors in the future are needed.

Based on the results of this study, efforts to develop and apply intervention programs to public health system management that can improve positive subjective health perceptions in patients with prediabetes/diabetes should be undertaken for the prevention and continuous management of diseases in patients with prediabetes/diabetes.

## Figures and Tables

**Figure 1 ijerph-19-07900-f001:**
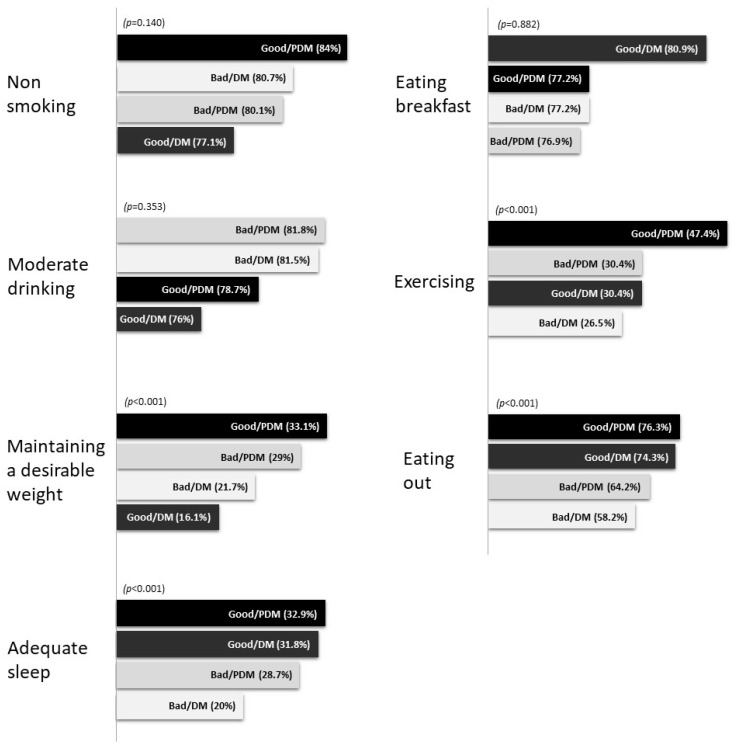
Health behavior practice rates between groups.

**Table 1 ijerph-19-07900-t001:** Chracteristics of subjects.

		Good/PDM		Bad/PDM		Good/DM		Bad/DM		Total		*χ*^2^/*F*	*p*
		*n*	%	*n*	%	*n*	%	*n*	%	*n*	%		
Sex	M	524	53.1	312	28.6	88	7.8	131	10.5	1055	100.0	35.061	<0.001
	F	599	43.5	592	39.8	63	4.6	176	12.1	1430	100.0		
	Total	1123	48.1	904	34.5	151	6.1	307	11.3	2485	100.0		
Weight		65.3 ± 0.4 ^(^^1)^		63.4 ± 0.5		68.7 ± 1.1		67.3 ± 1.0		64.0 ± 0.1		10.36 ^(2)^	<0.001
BMI		24.3 ± 0.1		24.8 ± 0.1		26.0 ± 0.3		26.4 ± 0.3		24.2 ± 0.1		26.06 ^(2)^	<0.001
Age	40–59 year	558	58.7	265	27.6	52	5.5	84	8.2	959	100.0	80.523	<0.001
	>60 year	565	38.1	639	40.9	99	6.7	223	14.3	1526	100.0		
	Total	1123	48.1	904	34.5	151	6.1	307	11.3	2485	100.0		
Marital status	Single	28	34.7	38	47.5	3	3.2	11	14.6	80	100.0	51.754	<0.001
	Married (living together)	911	52.9	602	31.0	121	6.1	202	10.0	1836	100.0		
	Married (living apart)	184	33.5	264	44.4	27	6.5	94	15.6	569	100.0		
	Total	1123	48.1	904	34.5	151	6.1	307	11.3	2485	100.0		
Household income	Q1	161	22.9	356	51.5	43	6.7	144	18.9	704	100.0	193.434	<0.001
	Q2	268	48.1	229	34.1	32	5.1	73	12.7	602	100.0		
	Q3	319	58.7	170	28.5	36	5.2	49	7.6	574	100.0		
	Q4	372	62.3	145	24.6	38	6.7	39	6.4	594	100.0		
	Total	1120	48.2	900	34.5	149	6.0	305	11.3	2474	100.0		
Employment	Yes	736	57.3	412	28.2	96	6.7	116	7.8	1360	100.0	103.950	<0.001
	No	386	34.5	489	43.8	55	5.2	190	16.5	1120	100.0		
	Total	1122	48.1	901	34.5	151	6.1	306	11.3	2480	100.0		
Education level	Elementary school	219	24.8	440	50.5	51	6.4	149	18.3	859	100.0	264.604	<0.001
	Junior high school	124	35.3	136	41.1	28	9.4	57	14.3	345	100.0		
	High school	379	55.7	209	29.7	38	4.8	75	9.8	701	100.0		
	University	400	70.6	115	19.5	34	5.7	24	4.2	573	100.0		
	Total	1122	48.1	900	34.4	151	6.1	305	11.3	2478	100.0		
Menopause	No	156	62.4	72	27.5	5	1.8	18	8.2	251	100.0	44.015	<0.001
	Yes	432	38.1	508	43.2	56	5.4	155	13.3	1151	100.0		
	Total	588	43.5	580	39.8	61	4.6	173	12.2	1402	100.0		

^(1)^ Mean ± SE ^(2)^ F-statistics.

**Table 2 ijerph-19-07900-t002:** Health behavior practice rate between groups.

	Good/PDM		Bad/PDM		Good/DM		Bad/DM		Total		*χ* ^2^	*p*
	*n*	% ^(^^1)^	*n*	%	*n*	%	*n*	%	*n*	%		
Non-smoking	980	84.0	756	80.1	119	77.1	248	80.7	2103	81.9	5.477	0.140
Moderate drinking	896	78.7	760	81.8	116	76.0	255	81.5	2027	79.9	3.259	0.353
Maintaining a desirable weight	389	33.1	268	29.0	28	16.1	64	21.7	749	29.4	24.155	<0.001
Adequate sleep	359	32.9	263	28.7	45	31.8	71	20.0	738	29.9	15.694	0.001
Eating breakfast	901	77.2	722	76.9	126	80.9	244	77.2	1993	77.3	0.915	0.822
Exercising	520	47.4	263	30.4	43	30.4	80	26.5	906	38.1	62.357	<0.001
Eating out	713	58.2	725	76.3	104	64.2	241	74.3	1783	66.6	58.032	<0.001

^(1)^ The percentage of health behavior practice.

**Table 3 ijerph-19-07900-t003:** Blood pressure and blood indices between groups.

	Good/PDM	Bad/PDM	Good/DM	Bad/DM	*F*	*p*
SBP	120.88 ± 0.52	124.44 ± 0.62	126.4 ± 1.27	128.88 ± 1.4	15.53	<0.001
DBP	77.06 ± 0.34	76.21 ± 0.42	77.92 ± 1.07	75.84 ± 0.83	1.73	0.160
TC	202.61 ± 1.28	193.08 ± 1.65	194.49 ± 4.39	181.6 ± 2.89	18.62	<0.001
HDL-C	50.18 ± 0.4	47.88 ± 0.43	43.87 ± 1.04	44.8 ± 0.86	19.14	<0.001
LDL-C	122.43 ± 2.80	120.39 ± 3.65	126.28 ± 5.76	105.44 ± 4.95	3.43	0.017
CVD score	10.75 ± 0.15	12.24 ± 0.16	12.59 ± 0.33	13.41 ± 0.26	35.53	<0.001
TG	141.99 ± 3.79	155.64 ± 5.82	199.56 ± 15.43	197.14 ± 15.44	9.16	<0.001
Glucose	100.55 ± 0.44	101.88 ± 0.58	147.78 ± 4.17	150.54 ± 3.25	111.59	<0.001

## Data Availability

Not applicable.

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
