# Peer review of "Effects of Subjective Health Perception on Health Behavior and Cardiovascular Disease Risk Factors in Patients with Prediabetes and Diabetes"

_ijerph, 2022, doi:10.3390/ijerph19137900_

Round 1

Reviewer 1 Report

Kwak S et al presented a manuscript entitled "Effects of subjective health perception on health behavior and cardiovascular disease risk factors in patients with prediabetes and diabetes. This article is a second analysis of raw data from a Korean survey that aimed to understand whether health behavior adherence could be related to subjective health perception or subjective health status (SHS) in patients with prediabetes and diabetes. SHP was assessed on a 5-point Likert scale (very good, good, and poor to very poor). Seven health-related behaviors were analyzed. Inclusion of patients with prediabetes and diabetes was defined by their HbA1c levels (>5.6 and >6.5%) but was not reported in the outcome tables. 

Four groups were cross-classified: good, poor, prediabetes, diabetes. Statistical methods and analyses are appropriate for these analyses. A short, focused conclusion is needed after the final summary for the public health.

There are several ways to improve the manuscript. 

1/ A table for health behaviors is not presented but appears to be replaced by 7 figures (Figure1 A-G). This paragraph is so confusing that it should be revised in detail with a complete table of results. The 7 figures are not presented in the results section. Therefore, this section is very confusing. No significant results are presented for eating breakfast (Fig. 1E).

2/ BMI and body weight are not presented for clinical characteristics of patients. Obesity is essential for conducting diet and lifestyle interventions. These important parameters should be analyzed and discussed. Duration of diabetes and HbA1c levels should also be analyzed.

3/ For the assessment of dyslipidemia, TC, HDL-C and TG are presented but not LDL-C, what is the reason? 

4/ the reference part is made of old references 2000-2015, only one 2021. The most recent ones should be presented. For example, the most recent article found with a multicenter clinical research design "PREVIEW" by Zhu R in Diabetologia 2022 should be cited because the results for diabetes prevention are different for older adults and women in a similar way to the present analysis. Disease perception and self-management practices are also important in achieving patient adherence to diabetes improvement (Ngetick et al 2022 Arch Public Health). 

Minor points.

Table 1 Bed should be Bad

Table 2 Glucose is FPG (mean fasting blood glucose).

line268 FRS should be defined

Author Response

Thanks for your comments.

  1. A table for health behaviors is not presented but appears to be replaced by 7 figures (Figure1 A-G). This paragraph is so confusing that it should be revised in detail with a complete table of results. The 7 figures are not presented in the results section. Therefore, this section is very confusing. No significant results are presented for eating breakfast (Fig. 1E).

=> You didn't mention the significance of breakfast. Tables are inserted, and the meanings of the corresponding figures are written.

  1. BMI and body weight are not presented for clinical characteristics of patients. Obesity is essential for conducting diet and lifestyle interventions. These important parameters should be analyzed and discussed. Duration of diabetes and HbA1c levels should also be analyzed.

=> BMI and weight was added to the result table.

=> PreDiabetes were also included as participants, so the analysis on Duration of Diabetes was not carried out. Similarly, HbA1c levels used to differentiate between PDM and DM were not used to analyze.

  1. For the assessment of dyslipidemia, TC, HDL-C and TG are presented but not LDL-C, what is the reason?

=> We added.

  1. the reference part is made of old references 2000-2015, only one 2021. The most recent ones should be presented. For example, the most recent article found with a multicenter clinical research design "PREVIEW" by Zhu R in Diabetologia 2022 should be cited because the results for diabetes prevention are different for older adults and women in a similar way to the present analysis. Disease perception and self-management practices are also important in achieving patient adherence to diabetes improvement (Ngetick et al 2022 Arch Public Health).

We added like below.

[Reference]

Zhu R, Craciun I, Bernhards-Werge J, Jalo E, Poppitt SD, Silvestre MP, Huttunen-Lenz M, McNarry MA, Stratton G, Handjiev S, Handjieva-Darlenska T, Navas-Carretero S, Sundvall J, Adam TC, Drummen M, Simpson EJ, Macdonald IA, Brand-Miller J, Muirhead R, Lam T, Vestentoft PS, Færch K, Martinez JA, Fogelholm M, Raben A. Age- and sex-specific effects of a long-term lifestyle intervention on body weight and cardiometabolic health markers in adults with prediabetes: results from the diabetes prevention study PREVIEW. Diabetologia. 2022 May 25. doi: 10.1007/s00125-022-05716-3. Epub ahead of print. PMID: 35610522.

Ngetich E, Pateekhum C, Hashmi A, Nadal IP, Pinyopornpanish K, English M, Quansri O, Wichit N, Kinra S, Angkurawaranon C. Illness perceptions, self-care practices, and glycemic control among type 2 diabetes patients in Chiang Mai, Thailand. Arch Public Health. 2022 May 7;80(1):134. doi: 10.1186/s13690-022-00888-1. PMID: 35524335; PMCID: PMC9078014.

Reviewer 2 Report

The authors present an interesting study examining the idea of health perception on health behaviours in the context of those living with variations of diabetes. Specifically, the authors utilise data from widely distributed survey to initially categorise those with which meet the inclusion criteria into one of four analyses groups; good/bad prediabetes/diabetes, before scrutinising the resultant data to determine specific perceptions and behaviours within those populations with respect to their physiological condition. A number of interesting trends across physiological condition, age, gender, and many others are drawn, with the overall arching finding being the better the subjective health status, the lower the risk of diabetes-associated cardiovascular disease. Overall, studies like this have value in the area of preventative medicine, and the findings here are interesting in that regard.

In reviewing the manuscript however, I had a number of concerns. The following should be addressed by the authors when preparing a suitable revision.   

1.       The writing of the manuscript needs revision/improvement. There are several instances where improvements could be made to the formatting, grammar, and language. The points being made, for the most part, or decipherable, but the manuscript is not to publication standard in its current form. The authors must review this aspect of the manuscript in advance of any resubmission.  

2.       Figure 1 and its subfigures require substantial revisions. The format of the graph is not clear. The labelling/text is hard to read/insufficient. The statistical comparisons are not clearly labelled/present. The figures legends could be more informative. Overall, this aspect needs substantial revisions.

3.       For clarity, the authors mention how data from seemingly two different studies; the 7th and 8th Korea National Health and Nutrition Examination Surveys, was utilised for this study. Is it possible that data appeared twice in the analyses with individuals having submitted data to each, or was this accounted for in some way?

4.       Can the authors clarify why ‘avoiding snacks’ was removed from the analyses?

5.       The discussion can be improved in terms of how the data produced is interpreted. At times the writing comes across as someone biased in how certain trends are linked to health. This is exemplary in the opening line whereby the authors state ‘modern lifestyles negatively impact their health’ which is incredibly broad in scope, and simply unfair/inaccurate to say. This trend follows suit throughout the discussion, and the authors need to re-evaluate their approach to this piece of the article.  

Author Response

Thanks for your comments.

  1. The writing of the manuscript needs revision/improvement. There are several instances where improvements could be made to the formatting, grammar, and language. The points being made, for the most part, or decipherable, but the manuscript is not to publication standard in its current form. The authors must review this aspect of the manuscript in advance of any resubmission.

  1. Figure 1 and its subfigures require substantial revisions. The format of the graph is not clear. The labelling/text is hard to read/insufficient. The statistical comparisons are not clearly labelled/present. The figures legends could be more informative. Overall, this aspect needs substantial revisions.

=>The graph has been modified.

  1. For clarity, the authors mention how data from seemingly two different studies; the 7th and 8th Korea National Health and Nutrition Examination Surveys, was utilised for this study. Is it possible that data appeared twice in the analyses with individuals having submitted data to each, or was this accounted for in some way?

=> The same subject may appear twice in the 7th and 8th orders, but there were no duplicate data in the data used in this study. In addition, the guideline for data use states that data(7th and 8th) can be integrated by using an integrated weight for each survey.

  1. Can the authors clarify why ‘avoiding snacks’ was removed from the analyses?

=>The survey on “avoiding snacks” was investigated up to the 4th round of the National Health and Nutrition Examination Survey, but “avoiding snacks” was not investigated in the 7th and 8th surveys used in this study.

  1. The discussion can be improved in terms of how the data produced is interpreted. At times the writing comes across as someone biased in how certain trends are linked to health. This is exemplary in the opening line whereby the authors state ‘modern lifestyles negatively impact their health’ which is incredibly broad in scope, and simply unfair/inaccurate to say. This trend follows suit throughout the discussion, and the authors need to re-evaluate their approach to this piece of the article.

=> We deleted biased phrases related to health

Round 2

Reviewer 1 Report

The authors have taken in considerations our remarks. The manuscript presents today some data LDL in particular which are essentials. Hba1C for the diabetes patients is a criteria interesting for the evolution of the disease. 

Author Response

Thanks for your comments.

Reviewer 2 Report

The authors have addressed the majority of my comments, and as a result the manuscript is much improved. I still feel that Figure 1 needs improvements made. The overall style of the figure should be revised such that the labels are easier to read, and the statistical comparisons being made between groups is much clearer. 

Author Response

Thanks for.

Based on your comments, the graph has been revised again.

Please review.
